# Influence of Soil and Water Conservation Measures on Soil Microbial Communities in a Citrus Orchard of Southeast China

**DOI:** 10.3390/microorganisms9020319

**Published:** 2021-02-04

**Authors:** Bobo Wu, Peng Wang, Adam T. Devlin, Shengsheng Xiao, Wang Shu, Hua Zhang, Mingjun Ding

**Affiliations:** 1School of Geography and Environment, Jiangxi Normal University, Nanchang 330022, China; wbb15279839084@163.com (B.W.); atdevlin@jxnu.edu.cn (A.T.D.); shuwang0717@jxnu.edu.cn (W.S.); zhangh2013@jxnu.edu.cn (H.Z.); dingmingjun1128@163.com (M.D.); 2Key Laboratory of Poyang Lake Wetland and Watershed Research, Ministry of Education, Jiangxi Normal University, Nanchang 330022, China; 3Jiangxi Institute of Soil and Water Conservation, Nanchang 330029, China; xss19811213@163.com

**Keywords:** soil and water conservation measures, high-throughput sequencing, soil bacterial–fungal, community and network, diversity

## Abstract

Soil microbes play a crucial role in ecosystem function. Here, the effects of soil and water conservation measures on soil microbial community structures, biodiversity, and co-occurrence networks are investigated and compared. We sampled soils at three different depths (0–10 cm, 10–20 cm and 20–40 cm) in a citrus orchard that uses long-term soil and water conservation measures, which includes Bermuda grass strip intercropping (BS), Bermuda grass full coverage (BF), Radish–soybean crop rotation strip intercropping (RS) and clear tillage orchards (CT). Results demonstrated that BS and BF yields a significant increase in bacterial richness and diversity of fungal in soils, while BF contains more beneficial microbial taxa, especially those with degrading and nutrient cycling capabilities. Microbial community structures differed significantly among the applied measures. In addition, co-occurrence networks under BS, BF and RS were more complex and robust than that of CT, and the stability of the network in BF was the highest. Microbial interactive stability and potential interactions in bacterial networks were stronger than those of fungi. The distribution of dominant phyla showed that *Chloroflexi* and *Ascomycota* dominated the different soil and water conservation measures. *Proteobacteria* and *Ascomycota* are revealed to be keystone species in bacterial networks and fungal networks, respectively, while *Proteobacteria* was the keystone species in microbial networks. Though the relative abundance of *Chloroflexi* turned out to have increased among the four measures, the relative abundance for *Proteobacteria*, *Acidobacteria* and *Actinobacteria* all decreased along the soil profile, with *Acidobacteria* under BS to be an exception. Soils under BS and BF had higher total nitrogen, microbial biomass carbon and organic carbon than CT and RS. Organic carbon(C) and total nitrogen(N) in soil were the major drivers of these bacterial community patterns, while there was no significant correlation between them and fungi. Overall, BF increases soil nutrients and microbial diversity, and also promotes ecological stability and interrelations among microbial taxa that collectively improve soil quality in the citrus orchard studied. Therefore, we recommended BF to be an ideal application for citrus orchards of southeast China.

## 1. Introduction

Soil microbial communities carry out key processes in nutrient cycling, ecosystem stabilizing, and maintenance of soil structure. The diversity of such communities is an important indicator of soil quality that can reflect subtle changes and provide information for the evaluation of soil function [1]. Bacteria and fungi are two main types of soil microorganisms, usually accounting for 90% of total soil microorganisms. Each plays different roles in soil biogeochemical processes and are of great significance for the sustainability of agricultural production systems [2]. Soil fungi and bacteria exhibit different community dynamics, so it is essential to address fungal and bacterial communities simultaneously to understand overall soil ecosystem processes.

Soil erosion is a severe ecological problem, especially in mountainous terrain, that seriously restricts sustainable forest development and the construction of ecological civilizations [3]. The combination of steep terrain and abundant rainfall can lead to serious soil erosion during the initial stages of orchard development, which will affect subsequent sustainable maintenance of orchards [4]. Additionally, the strong interference of large-scale mechanized soil excavation in orchards can lead to an imbalance of soil nutrient cycles and a decline in crop productivity [5]. Mountainous regions are common in the red soil area of Southern China, covering about 60.6% of the land area. With the impact of rapid population growth and the development of the fruit industry, a large number of secondary forests in sloping terrain regions have been transformed into orchards [6]. Exploring new soil and water conservation measures is thus critical for the sustainable development of the orchard industry in this region.

Soil and water conservation measures are generally associated with physical and chemical changes in soil characteristics, such as changes in soil nutrient availability [7,8]. The impacts of soil and water conservation measures on soil moisture, soil structure, soil nutrition and crop yield have been previously discussed, [9,10], but these studies have not thoroughly investigated the influences on soil micro-ecological environments and the evolution of microbial communities in soil, and the detailed effects of continuous measures on soil microbial communities and the link between these effects and soil environments remain unclear. In addition, changes in physical and chemical properties of soil due to conservation measures may lead to shifts in the composition and function of microbial communities, as different groups of microorganisms vary in their ability to adapt to various soil environmental conditions. Most studies have reported that, compared with RS and CT, BF increased soil organic matter content and microbial diversity, and is an effective strategy to improve soil microenvironment [11,12]. RS increased the number of bacteria in the soil but did not affect the richness or diversity of bacteria [13]. Chen et al. [14] observed that CT easily leads to the decrease of soil organic matter, the decrease of bacterial diversity, the premature senility of trees and the decline of fruit quality. In contrast, Dong et al. [15] believed that CT could increase soil carbon storage and nutrient content, stabilize soil structure, and thus enhance soil microbial activity. Sanaullah et al. [16] found that the leguminous plants under RS reduced the pressure of pests and diseases, increased the nitrogen content in soil, and thus led to the enhancement of soil microbial activity and diversity. Thus, there are few and sometimes contradictory studies about the changes in microbial community structure and their drivers (e.g., soil physicochemical properties), which limits our understanding of soil biogeochemical processes under various soil and water conservation measures. In addition, Banerjee et al. [17] have shown that microbial co-occurrence networks and keystone species in networks vary among different soil nutrient levels as well as by spatial habitats. Exploring co-occurrence patterns between soil microorganisms can help identify potential biotic interactions and analyze niche sharing or competition between bacterial and fungal taxa at the aggregate scale [18].

In the present study, four soil and water conservation measures were applied in a citrus orchard located in southeast China during the time period of 2001 to 2019. Soil physicochemical properties and microbial (bacterial and fungal) community structures were analyzed under the different treatments. The associations between microbial communities were also identified using network analysis. To accomplish this, we hypothesized that various mulching practices, especially BF, may improve soil nutrient levels and elicit different responses from microbial communities, while altering the interrelations among microorganisms. We specifically investigated: (i) the changes in soil microbial communities in a citrus orchard that have experienced long-term soil and water conservation measures, (ii) the relationship between microbial communities and soil properties, e.g., soil organic carbon and total nitrogen and (iii) the network complexity, keystone taxa and ecological stability in the microbiome. The results presented here will help quantify the influence of soil and water conservation measures for citrus orchard soil microbes in other parts of southeast China and other related biomes worldwide and provide a theoretical foundation for evaluating sustainable management and conservation policies in soil ecosystem systems.

## 2. Materials and Methods

### 2.1. Study Area, Sampling and Physiochemical Analysis

Our study area lies in the Jiangxi Provincial Eco-Science Park of Soil and Water Conservation (29°16′–29°17′ N, 115°42′–115°43′ E), which is located in De’an County, Jiangxi Province, China (Appendix A). This region is characterized by a subtropical humid monsoon climate with a mean annual precipitation of 1397.3 mm. The area has an average annual air temperature of 16.7 °C, with the maximum and minimum temperatures typically occurring in July and January, respectively. The annual sunshine hours are 1650~2100 h, and the average annual frost-free period is 149 days [19]. The field soil is red soil (fine, kaolinitic, thermic, Udults) and the soil thickness ranges from 0.5 to 1.5 m, most of which exists in hilly terrain. The vegetation status types are mainly natural secondary, semi-secondary and artificial trees and their associated shrubs and ground cover vegetation. The Eco-Science Park of Soil and Water Conservation is located in the central region of red soil distribution, which belongs to the second-class soil erosion type area in China and is typically representative of Jiangxi province and southern red soil hilly area.

The experimental area is a citrus orchard soil and water conservation measures experimental area. The citrus orchard was planted in 2001 according to an initial density of 1335 plants/hm^2^. Four treatments of four runoff plots were studied: Bermuda grass strip intercropping (BS), Bermuda grass full coverage (BF), Radish–soybean crop rotation strip intercropping (RS) and clear tillage orchards (CT). Four runoff plots with slopes of 10°. Each runoff plot was 2 m wide by 10 m long, and had concrete borders that extended 20 cm.

In May 2019, we sample a mixture of five sample points and three replicates from five randomly selected sites in each runoff plot. At each site, soil samples were taken vertically at the following depth intervals (cm): 0–10, 10–20 and 20–40. After visible stones and plant residues were removed by hand, the soil samples were homogenized and passed through a 2 mm sieve. Each soil sample was divided into two parts, placed under ice, and shipped to the laboratory within 4 h. One sub-sample was air-dried and used to determine the physiochemical properties of the soil. The second sub-sample was immediately frozen at −80 °C for DNA extraction and microbial community analysis. Due to the high cost of sequencing, the samples used for high-throughput sequencing were mixed into one sample from each of the three repeated samples.

### 2.2. DNA Extraction and 16S rRNA Gene Amplicon Sequencing

Total phosphorus (TP) concentrations of the soil samples were measured using NaOH and H_2_SO_4_ and reaction with a molybdenum antimony antichromator for a colorimetric analysis. The soil total nitrogen (TN) was simultaneously measured with an elemental analyzer (Vario Macro, Elementar, Germany). The pH was measured using a pH meter after mixing the soil with water (1:5 *w*/*v*) for 30 min. Soil organic carbon (SOC) level was assessed by wet oxidation with dichromate oxidation and titration with ferrous ammonium sulfate. Particle size distribution was analyzed using laser diffraction per the method of Eshel et al. [20]. The concentration of dissolved organic carbon (DOC) was analyzed using titration after wet oxidation with H_2_SO_4_ and K_2_Cr_2_O_7_. Microbial biomass carbon (MBC) content was determined using chloroform fumigation-extraction.

Microbial DNA was extracted using E.Z.N.A.^®^ Soil DNA Kit (Omega Bio-tek, Norcross, GA, USA) according to the manufacturer’s protocols. PCR was performed with the Takara ExTaq PCR kit (Takara Shuzo, Osaka, Japan). The V3-V4 regions of the bacterial 16S rRNA gene were amplified using the following PCR conditions: 95 °C for 2 min, followed by 25 cycles of 95 °C for 30 s, 55 °C for 30 s, and 72 °C for 30 s, and a final extension at 72 °C for 10 min. The following PCR primers were used for this amplification: 338F (5′-ACTCCTACGGGAGGCAGCAG-3′) and 806R (5′-GGACTACHVGGGTWTCTAAT-3′). The region of the fungal ITS ribosomal gene was amplified using the following PCR conditions: 95 °C for 2 min, followed by 25 cycles of 95 °C for 30 s, 55 °C for 30 s, and 72 °C for 30 s, and a final extension at 72 °C for 10 min. The primers ITS1F (5′-CTTGGTCATTTAGAGGAAGTAA-3′) and ITS2R (5′-GCTGCGTTCTTCATCGATGC-3′) were used. The amplicons were extracted from 2% agarose gels and purified using the AxyPrep DNA Gel Extraction Kit (Axygen Biosciences, Union City, CA, USA) and quantified using QuantiFluor™-ST (Promega, Madison, WI, USA) according to the manufacturer’s instructions. The above procedures were sequenced using the Illumina MiSeq platform by the Shanghai Majorbio Bio-pharm Technology Co., LBS., Shanghai, China. Raw reads have been deposited in the NCBI Sequence Read Archive database (access number: SUB7239364).

Quantitative PCR (qPCR) was used to determine bacterial 16S and fungal ITS target abundance. Primer set 338F/806R/ITS1F/ITS2R was used to determine the bacterial and fungal abundance by quantitative real-time PCR (qPCR) on a LineGene9600plus Optical Real-Time Detection System. The 20 μL qPCR reaction mixture contained ChamQSYBRColorqPCRMasterMix (2X)10 μL, 0.5 μlPCR forward and reverse primer, 2 μL DNA template, and 4.5 μL double distilled water (ddH_2_O). Three parallel samples were set for each sample. Quantitative real-time PCR parameters were as follows: hold at 95 °C for 5 min, then 40 cycles with 95 °C for 30 s, 56 ℃ for 30 s and 72 °C for 40 s. Plasmid DNA containing fragments of fungal ITS gene were used as standards. The potential for PCR inhibition in the extracts was tested by qPCR of dilutions; absence of inhibition was found. Melting curve analysis showed excellent specificity of the qPCR with an efficiency of 95% (r^2^ = 0.9995).

### 2.3. Statistical Analyses

Operational Taxonomic Units (OTUs) were clustered with a 97% similarity cutoff using UPARSE (version 7.1) and chimeric sequences were identified and removed using UCHIME. The phylogenetic affiliation of each 16S rRNA gene sequence was analyzed by RDP Classifier (Release 11.3) against the Silva (Release 119) 16S rRNA database using a confidence threshold of 70%. Dilution curve analysis was performed based on OUT, We evaluated the alpha-diversity through the Chao1 richness index and the Shannon diversity index. All the data were tested for normality (Shapiro–Wilkes test). Variables that were not normally distributed were log-transformed to normality. Pearson correlations and one-way analyses of variance (ANOVA) with Fisher’s least significant difference (LSD) posthoc tests were performed using SPSS Statistics v20. The LSD method was used for multiple comparison, and Pearson correlation analysis was used for correlation analysis (with a significance level of *p* ≤ 0.05 considered as a significant difference). A Venn diagram was constructed to reflect the number of common and unique OTUs under different soil and water conservation measures. A Partial Least Squares Discriminant Analysis (PLS-DA) was carried out on OTU data using Bray–Curtis distance matrices to examine the difference in microbial community between soil samples. Redundancy analysis (RDA) was employed to explore the relationship between environmental parameters and the microbial community.

Network analysis (based on phyla level) was performed to identify the interrelations between microbial taxa, using Cytoscape version 3.4.0 combined with the CONET plug-in (http://psbweb05.psb.ugent.be/conet) [21]. Cytoscape (version 3.4.0) was used for network visualization and modularization analysis to determine associations (positive and negative correlations) between bacterial community members [22]. To highlight the most important interactions, only strong positive (*r* > 0.8) and strong negative (*r* < −0.8) relationships were shown in the network diagrams [17]. The Network Analyzer tool was used to calculate the network topology parameters, such as average clustering coefficient, geodesic distance, connectivity and network diameter. Phyla with the highest centrality scores were considered keystone species [23]. The classification identifier for each OTU was assigned at the category level. The resulting original OTU table (one with taxonomic abundance) was used as the input matrix. The network was built according to the guidelines provided on the CONET website (http://psbweb05.psb.ugent.be/conet/tutorial4.php). Parameters were set as follows: at least 30 sequences are preprocessed and filtered for each OTU, and there are four similarity measures (Spearman, Pearson, Kullbackleibler and Bray-Curtis) and automatic threshold setting. Error detection rate (FDR) correction was set to 0.05 (*p* < 0.05).

## 3. Results

### 3.1. Soil Physicochemical Properties and Microbial Community Quantity

The soil physicochemical properties varied with depth. TN, SOC and MBC significantly increased in BS and BF (especially BF). Table 1 shows the average TN, SOC and MBC under BF in the surface, middle and subsurface soil was higher than that under BS, RS and CT, respectively. However, CT contained significantly higher TP than the three other treatments. Additionally, the pH, GC, DOC and MBC were found to be different between each vertical layer. However, there were no significant differences in MBC among the three layers.

Bacterial and fungal abundance increased significantly in response to BF, indicating that BF increased the number of microbes in the citrus orchard soil. The average 16S rRNA and ITS rRNA gene copy number under BF in the surface, middle and subsurface soil was the highest (Table 1). In particular, the gene copy numbers at the surface were significantly higher than that in the subsurface in BS and BF. Interestingly, the TN and SOC decreased dramatically along the entire profiles and were significantly positively correlated with 16S rRNA (Table 2) and ITS rRNA (Table 3) gene copy number in BF or BS, respectively. However, the gene copy numbers at the surface were clearly less than that in the subsurface and increased with decreasing TN and SOC under CT. Additionally, TP increased dramatically along soil profiles and was significantly positively correlated with gene copy number under CT.

### 3.2. Fine Scale Spatial Distribution Patterns of Soil Microbiomes

A total of 623,653 and 796,735 high quality bacterial and fungal sequences were obtained after quality filtering of the 12 soil samples, grouping into 456,828 and 637,404 OTUs when using the 97% sequence similarity cutoff, respectively. We used the mean numbers of 3531 (bacteria) and 1135 (fungi) OTUs per sample to evaluate the soil bacterial and fungal richness, diversity, and evenness. The overlapped and unique OTUs are illustrated by a Venn diagram (Appendix A). The four communities shared 1217 and 51 OTUs, which accounted for 37.67% and 4.49% of the total bacterial and fungal OTUs, respectively. Among all different treatments, BF contained the most microbial species and the evenness index was the highest.

BF and BS significantly increased fungal alpha diversity and bacterial richness index (Chao1) compared with CT in surface and middle soil (Figure 1). Figure 1 and Figure 2 show that the microbial diversities of BF were significantly higher in surface layers than deep layers and the Shannon index for the surface layer of soil in BF was the highest. Appendix A shows the correlation between the diversity index and environmental characteristics. The Shannon index of bacteria was significantly positively correlated with TN, pH and SOC, while those contents effect on the bacterial richness index (Chao1) was not significant. Similarly, the Shannon and Chao1 index of fungi was significantly positively correlated with MBC.

There were significant differences in soil microbial communities under four soil and water conservation measures. PLS-DA clearly demonstrated variations among the different soil samples, with the first two axes explaining 13.72% and 11.8% of the total variation for the bacteria and 15.13% and 10.33% for the fungi, respectively (Figure 3). Due to a limited sampling number, there is a limited explanation for the observed variability. Additionally, we observed significant differences (especially BF) in soil microbial communities between the surface and deep layers and these differences (especially CT) were larger for bacteria than fungi.

### 3.3. Predominant Microbial Taxa of Soil Microbial Communities

A total of 3531 bacterial OTUs included here belonged to 34 taxonomic groups in the soil samples. All OTUs were assigned to 1241 species, 616 genera, 368 families, 230 orders, 93 classes, and 34 phyla. The majority of bacterial sequences belonged to *Chloroflexi*, and the second-largest phylum was *Acidobacteria*, followed by *Proteobacteria* and *Actinobacteria* (Figure 4). In addition, A total of 1135 fungal OTUs included here belonged to 15 taxonomic groups in the soil samples. All OTUs were assigned to 308 species, 229 genera, 146 families, 73 orders, 36 classes, and 15 phyla. The most abundant fungal phylum was *Ascomycota*, followed by *Mortierellomycota*, *Basidiomycota* and unclassified (Figure 5).

The distribution of phyla showed that *Chloroflexi* and *Ascomycota* dominated the different soil and water conservation measures, whereas several of the phyla showed different vertical distribution patterns. The relative abundance of *Chloroflexi* was the highest in CT, while that of *Acidobacteria* and *Proteobacteria* were lower. In surface soil samples, BF and CT revealed a higher abundance of sequences affiliated with *Proteobacteria*, while samples from BF and RS had higher numbers of sequences related to *Acidobacteria*. In subsurface soil samples, CT had greater abundances of sequences affiliated with *Chloroflexi* than BS, BF and RS, while that of *Acidobacteria* and *Proteobacteria* were lower. Though the relative abundance of *Chloroflexi* turned out to have increased among the four measures, the relative abundance for *Proteobacteria*, *Acidobacteria* and *Actinobacteria* all decreased along the soil profile, with *Acidobacteria* under BS being an exception. For fungi, the relative abundance of *Ascomycota* was the lowest among soil layers of BF compared to BS, RS and CT. Similarly, the relative abundance of *Mortierellomycota* in CT was the lowest. However, the relative abundance of *Basidiomycota* in BF was the highest, especially in subsurface soil.

Soil bacterial and fungal communities were found to be significantly different, with TN and SOC identified as the major determinants of bacterial community characteristics. RDA was used to determine the relationship between the main microbial phylum and environmental parameters (Figure 6). The first two axes of RDA explain 63.96% and 3.5% of the total variation in the bacterial data. The environmental parameters that contributed significantly to the bacterial community–environment relationship were TN (*p* = 0.009) and SOC (*p* = 0.014). *Acidobacteria*, *Proteobacteria* and *Actinobacteria* were significantly positively correlated with TN and SOC, while the most abundant phylum *Chloroflexi* was negatively (Figure 6a). Similarly, the first two axes of RDA explain 28.35% and 13.56% of the total variation in the fungal data. The results of RDA demonstrated that the TN (*p* = 0.646), TP (*p* = 0.378), pH (*p* = 0.346), MBC (*p* = 0.58) and DOC (*p* = 0.742) had stronger effects (a longer arrow) on the composition of the fungal communities than other properties, while there was no significant correlation between these and fungal communities (Figure 6b).

### 3.4. Interactions between Microbial Taxa in the Network

The network of bacterial and fungal communities revealed distinct co-occurrence patterns. These nodes belong to 21 bacteria phyla and 7 fungi phyla, with *Actinobacteria*, *Chloroflexi*, *Proteobacteria*, *Acidobacteria* and *Ascomycota*, *Basidiomycota*, and unclassified were mainly found (Figure 7). *Proteobacteria* and *Ascomycota* were the keystone species in bacterial networks and fungal networks, respectively. Furthermore, *Proteobacteria* was the keystone species in microbial networks and was not altered by different measures. In microbial networks, bacterial networks were larger, more connected and more modular than fungal networks (Table 4). fungal networks consistently had fewer negative correlations than bacterial networks. Compared to BS, RS and CT, the higher modularity and clustering coefficient resulted in the increasing interactions of bacterial–fungal taxa in BF. Compared to CT, BF increased the ratio of negative/positive (BF vs. CT = 0.979 vs. 0) in microbial networks, indicating that the microbial network under BF was more robust. In addition, Figure 8 shows that the BF bacterial network formed a much more complex network and harbored more keystone taxa than other networks. Compared to CT, the number of nodes in the bacterial network increased after implementing BS, BF and RS, among which BF was 47.6% higher than CT (Table 5). Although the node of CT was the lowest, the links were the largest. In particular, *Proteobacteria*, *Acidobacteria* and *Actinobacteria* were more closely linked, indicating that CT could promote the interaction (especially cooperation) between bacterial taxa. Figure 9 shows that BS significantly affected fungal community structure with a much more complex network as measured by the number of nodes and edges. In contrast, the CT network only consisted of seven nodes and five edges (Table 6).

## 4. Discussion

### 4.1. Soil Bacterial and Fungal Abundance and α-Diversity

BS and BF increased the abundance and alpha diversity of bacterial and fungal communities as compared to RS and CT. The soil using BS and BF had higher SOC and TN content (Table 1), which should be related to the return of C and N due to planting grass and contribute to the bacteria and fungi communities. Reckling et al. [24] and Cui et al. [11] also found similar results. Furthermore, the presence of grass roots and root exudates is a possible reason for the accumulation of SOC [25]. Table 2 and Table 3 show that TN and SOC were the primary drivers of the 16SrRNA and ITS rRNA gene copy number in surface soil microorganisms. Therefore, the higher surface soil TN and SOC of BS, BF and RS (especially BS and BF) can explain the nutrient sources required for the survival and reproduction of bacteria and fungi.

Moreover, BF and BS significantly increased fungal alpha diversity and bacterial richness index (Chao1) compared with CT in surface and middle soil. Appendix A shows that the Shannon index of bacteria was significantly positively correlated with TN, pH and SOC and the Shannon and Chao1 index of fungi was significantly positively correlated with MBC. Thus, BF and BS substantially increased SOC and MBC, and thus moderately perturbed the microbial community, decreased the competitive niche exclusion and selection mechanisms between populations, improved carbon source utilization and metabolic activity, and promoted soil microbial species diversity. Wu et al. [26] and Wang et al. [12] came to a similar conclusion that BF was a useful management practice for enhancing biological soil activity. In contrast, CT had a large perturbation to all soil layers and the homogenization of soil bacteria and fungi was significant. In this study, we found that TP increased dramatically along soil profiles and was significantly positively correlated with gene copy numbers under CT. Thus, the higher 16SrRNA and ITS rRNA gene copy numbers in subsurface soil may be explained by the increase of TP content under CT. However, the Shannon index in all soil layers of CT was the lowest. Our observation is in contrast with a previous study that shows CT had positive impacts on the diversity of the whole microbial community or specific groups of microorganisms in soils [15]. On the other hand, we observed that bacterial abundance, fungal abundance and diversity in the surface were lower in RS than BS and BF, which was different from the findings of Sanaullah [16]. In this manner, the frequent perturbation of the soil increased the amount of litter, produced more root exudates in the soil, enhanced the input of soil organic matter and reduced the C:N ratios, thus stimulating the specific functions of the microbial community in the surface soil and reducing the microbial diversity under RS.

In addition, BF generated higher differences in the microbial community in the soil profile (Figure 3), implying higher bacterial and fungal heterogeneity between these samples. The causes of the phenomenon may be the differences in distance between the samples and the roots of grass. Samples closer to grass roots have higher microbial diversity, because grass root exudates are beneficial to the improvement of microbial diversity [27]. Interestingly, these differences (especially CT) were larger for bacteria than fungi, indicating that the bacterial community in CT was more sensitive to soil depths and higher fungal stability between these samples. We speculate that this is due to a homogeneous soil environment for the development of fungi, which was shaped by implementing 19 years of clear tillage.

### 4.2. Soil Bacterial and Fungal Community Composition

Different soil and water conservation measures affected microbial diversity and altered the structure and composition of the bacterial community in the soil profiles, while TN and SOC emerged as the major drivers of bacterial community patterns. Changes in soil and water conservation measures have changed the original niche of soil microbes, therefore resulting in corresponding changes in the soil microbial community structure. Our results agree with Dong et al. [15], who showed that the soil physicochemical properties (especially TN and SOC) significantly affected bacterial community structure. TN was one of the most important factors for affecting the differences of bacterial communities in different planting systems and the importance of SOC in shaping other bacterial communities has also been reported [28]. Many previous studies of bacterial communities in the humid tropics also showed that *Chloroflexi* was the most abundant phyla, followed by *Acidobacteria*, *Proteobacteria* and *Actinobacteria* [29,30,31]. Due to abundance changes, these microbes will be actively involved in the microbe–nutrient interactions. Generally, *Chloroflexi* has a high demand for soil N but cannot fix N [32]. This could explain the high relative abundance of *Chloroflexi*.

In this study, we found that *Acidobacteria*, *Proteobacteria* and *Actinobacteria* were significantly positively correlated with TN and SOC, while the most abundant phylum *Chloroflexi* was negatively correlated (Figure 6a). The highest relative abundance of *Chloroflexi* in CT among the soil and water conservation measures may be attributed to a lower level of TN and SOC. *Proteobacteria* are more common in copiotrophic than in oligotrophic bacteria, widely distributed in plant litter and areas of root exudation, and play key roles in carbon cycles [33,34]. Due to higher N demands [35], copiotrophic taxa (including members of the *Proteobacteria* phyla) typically increased in relative abundance in the surface layer of BF, with oligotrophic taxa (mainly *Chloroflexi*) exhibiting an opposite pattern. Furthermore, *Chloroflexi* populations, which are known to metabolize carbohydrate and the common polysaccharide-degrading bacteria, appear to be widespread in soil bacteria, growing predominantly in orchard soils with low-oxygen availability and specialized in polysaccharide degradation [36]. Therefore, the relative abundance of *Chloroflexi* increased along the soil profile largely in the anaerobic environments of subsurface soil and inhibited the growth and reproduction of copiotrophic microbes.

We also observed that the relative abundance of *Proteobacteria*, *Acidobacteria* and *Actinobacteria* all decreased along the soil profile. The oxic habitat in soil may account for the increase of the relative abundances of *Proteobacteria* and *Actinobacteria*. In contrast, there was an increase in the relative abundance of *Acidobacteria*, a group under BS that is often considered to be oligotrophic with slower growth rates and, in all likelihood, the ability to degrade complex and difficult-to-degrade compounds and can use nitrite as an N source [37,38]. The reason for the observed trend under BS may be attributed to its high level of TN in middle and subsurface soil. As an important participant in the global cycling of carbon and nitrogen, *Acidobacteria* is beneficial to the sustained release of nutrient residues under BS.

Different soil and water conservation measures lead to changes in the composition of fungal communities and several of the phylum showed different vertical distribution patterns. Compared with bacteria, the fungal richness and diversity were lower and only 15 taxa were identified at the phylum level. Our results showed that *Ascomycota*, *Basidiomycota* and *Mortierellomycota* were the dominant fungal phyla, which was consistent with the previous studies in citrus orchard [39]. *Ascomycota* and *Basidiomycota* usually act as major decomposers of organic matter across different soil types because of variations in their nutritional niches [17,40]. Generally, *Ascomycota* largely dominates the active fungal community through its involvement in root exudation assimilation and soil organic matter degradation. In our study, compared to BS, RS and CT, the relative abundance of *Ascomycota* was the lowest among soil layers of BF. One plausible reason is that due to less vegetative cover in BS, RS and CT, bare land becomes more reflective and is often subjected to direct sunlight, which would lead to changes in the *Ascomycota* phyla. On the other hand, the relative abundance of *Basidiomycota* in BF was the highest (especially in subsurface soil) and RS was the lowest. *Basidiomycota* can be involved in C cycling by degrading organic substances [41]. Thus, we speculate that BF can provide a stable soil environment and *Basidiomycota* may participate in C cycling by degrading organic substances, whereas RS can easily damage the extensive branching mycelial network formed by *Basidiomycota* [42].

In addition, although we tested the effects of seven physicochemical properties on soil bacterial and fungal diversity, other abiotic factors such as soil temperature might also be important, and future studies should quantify their impacts on soil microorganisms. This could explain the fact that the fungal community structure of different soil and water conservation measures were significantly different but showed no significant correlation with the soil environmental parameters. However, this may indicate an important effect of vegetation type associated with soil and water conservation measures change on the structure of the soil fungal community. Vegetation was found to affect the soil microbial community structure [43]. Underground ecosystems with the roots of various plants may have a marked effect on soil microbial structures [44].

### 4.3. Interactions of the Soil Microbial Community in the Networks

BF increased the interactions of bacteria and fungi in the microbial network. A higher modularity and clustering coefficient indicated marginally higher modularity [45]. Our network analysis showed that compared to BS, RS and CT, the higher modularity and clustering coefficient yielded more bacterial–fungal links under BF, an outcome perhaps related to root exudates and plant residues. In the microbial networks, the co-occurrence of bacterial copiotrophic communities with the dominant fungi, particularly under BF, suggests their dependence on fungi for release of substrates from the recalcitrant fractions of Bermuda grass stems. Bacteria and fungi commonly co-participate in the decomposition of plant residues and the increased biomass stimulated by a greater supply of nutrients in larger fractions also provides more opportunities for different species to interact with one another [46]. Therefore, an abundance of Bermuda grass and citrus residues may enhance the myriad interrelations between synergistic and antagonistic bacterial and fungal groups, which result in functional association or phylogenetic clustering of closely related species was higher and the increasing interactions of bacterial–fungal in BF. In contrast, BS significantly affected the fungal community structure with a much more complex network (see Figure 8). It is possible that BS might have a higher degree of humification, which provides a hotbed for fungi. Increased fungal associations may also arise because of the functional dominance of fungi in the decomposition of cellulose and lignin. Furthermore, a complementarity/competition relationship exists between Bermuda grass and citrus at the plant development stage or under special environmental conditions. As the full covered practices were added, it may have increased the competition and decreased many trophic levels or resource cascades. As a result, the number of negative associations substantially increased between bacterial and fungal modules. In particular, compared to CT, BF increased the ratio of negative/positive in microbial networks, indicating that a possible competition for resources and common predators [47].

BF increased the microbial interactive stability in the network. Dundore-Arias [48] reported that due to niche overlap, the selective effect of soil carbon inputs may plausibly intensify nutrient competition among co-existing populations. The maximum carbon input intensified the nutrient competition among potential competitors of BF and the interaction of the soil bacterial community had the best stability. Furthermore, negative links might stabilize co-oscillation in communities and promote the stability of networks [49]. More generally, stability is promoted by limiting positive feedbacks and weakening ecological interactions. There were higher negative/positive ratios seen after implementing BF, which indicates that hosts can benefit from microbial competition when this competition dampens cooperative networks and increases stability. However, when CT with the lowest negative/positive ratio was perturbed by the external environment, the interaction network of the soil microbial community will transmit the environmental perturbation to the entire network in a short time and result in an unstable network structure. At the same time, this unstable network may lead to significant changes in the bacterial and fungal community involved in soil carbon and nitrogen and other nutrient cycles [50], which in turn affects vegetation growth. Therefore, the microbe (especially *Proteobacteria*, *Acidobacteria* and *Actinobacteria*) under CT with poor soil bottom nutrients can only respond to environmental pressures by reducing competition and strengthening cooperation.

Meanwhile, an increase in the number of modules indicates more niches, network modularity also can enhance the stability of the network under environmental disturbances [51]. Modularity measures the connectivity between nodes within their own modules that would not occur by chance [52]. Compared to BS, RS and CT, the higher modularity and clustering coefficient of BF resulted in higher stability of bacterial–fungal taxa. When BF was perturbed by the external environment, the bacterial and fungal taxa were more likely to cope with environmental perturbation through mutual cooperation. In contrast, the interaction network of BS and RS was easily perturbed by the external environment, the resource competition within the microbial community was weak in the soil, and the stability of the interaction of the soil microbial community was poor.

On the other hand, the BF bacterial network formed a much more complex (i.e., higher connectivity) and larger (i.e., more nodes and links) network (Figure 7). In this way, the increased complexity of the network structure in BF may lead to higher community stability with a mixed interaction, increase the breadth of the niche, strengthen the interconnections between different bacteria in the soil microbial food web, enhance the efficiency of resource transfer, and help it use soil nutrients more effectively; this result is similar to that reported by Mougi [53]. Additionally, although CT increased the number of positive and negative correlation links in the bacterial network, the Shannon index of all soil layers was the lowest. These stronger negative interactions between only a few species under CT exclude more species from the community and result in a loss of biodiversity. At the same time, these stronger interactions also decrease the stability of the bacterial communities, providing a mechanistic link between species interaction, biodiversity and stability [54].

Microbial interactive stability and potential interactions in bacterial networks are stronger than those of fungi. Indeed, network analysis identified fungal networks were characterized by properties that indicate lower stability under disturbance, such as lower modularity, while bacterial networks had properties that suggest higher stability. Furthermore, the finding that bacterial interactions were more active than fungi in soil environments may be due to their faster growth, high turnover rates and rapid responses to rhizospheric exudates. As a result, the most abundant OTUs were driving bacteria but not fungi in response to network reorganization. Furthermore, we found that the predominant taxa (especially *Proteobacteria*) drive network structure (highest centrality scores). Compared with other predominant bacterial phyla, *Proteobacteria* with copiotrophic advantages may demonstrate wider niche breadths and higher anti-interference capacities and play a dominant role in maintaining the stability of the soil bacterial community interaction network. Other less dominant phyla can be used as a diversified library to enhance the resilience of microbial and the resistance capacities to environmental perturbation [55]. While the relationships between bacterial phylogeny and function are complex, shifts in the abundance of indicator taxa might inform on the stability of these networks, and consequently on the response of soil bacterial communities to soil and water conservation measures.

## 5. Conclusions

In this study, we observed that the Bermuda grass strip intercropping (BS) and Bermuda grass full coverage (BF) soil treatment methods increased the abundance and alpha diversity of bacterial and fungal communities as compared to the Radish–soybean crop rotation strip intercropping (RS) and clear tillage orchards (CT) methods. Microbial community structures differed significantly among the applied measures. Microbial interactive stability and potential interactions in bacterial networks were stronger than those of fungal networks. *Chloroflexi* and *Ascomycota* dominated the different soil and water conservation measures, whereas several of the phyla showed different vertical distribution patterns. *Proteobacteria* and *Ascomycota* were keystone species in bacterial networks and fungal networks respectively, while *Proteobacteria* was the keystone species in microbial networks. Though the relative abundance of *Chloroflexi* increased among the four measures, the relative abundance of *Proteobacteria*, *Acidobacteria* and *Actinobacteria* all decreased with depth along the soil profile, with *Acidobacteria* under BS being an exception. Soils under BS and BF had higher TN, MBC and SOC than CT and RS. SOC and TN were the major drivers of these bacterial community patterns, while there was no significant correlation between bacteria and fungi.

In summary, Bermuda grass full coverage practices increase the diversity of soil microbial communities and the abundance of dominant species, and also promotes the stability and interrelations among microbial taxa that collectively improved soil quality in citrus orchard environments. The results of our study improve the understanding of soil microbial structure in typical subtropical humid agroecosystems, and these results can be used to predict the impact of soil and water conservation measures on the soil microbiome.

## Figures and Tables

**Figure 1 microorganisms-09-00319-f001:**
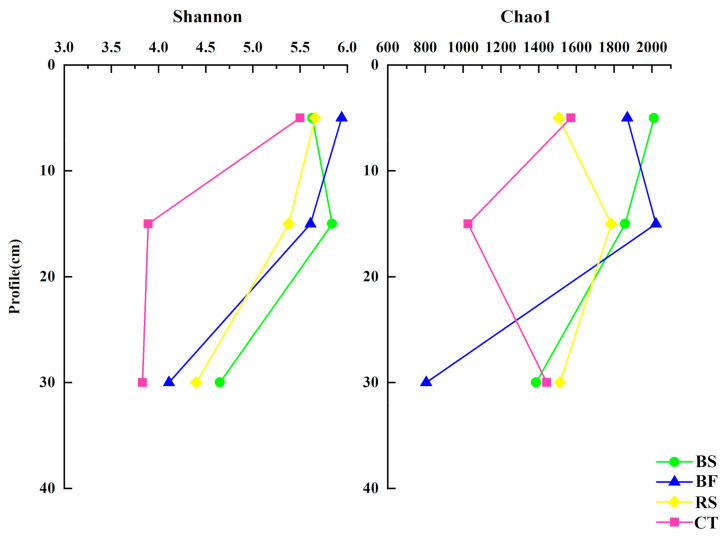
Diversity and richness in soil bacteria communities along the soil profile under different soil and water conservation measures.

**Figure 2 microorganisms-09-00319-f002:**
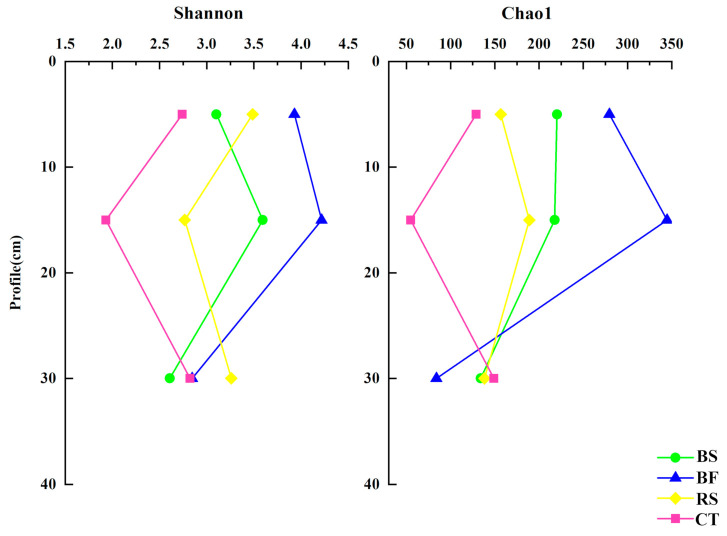
Diversity and richness in soil fungal communities along the soil profile under different soil and water conservation measures.

**Figure 3 microorganisms-09-00319-f003:**
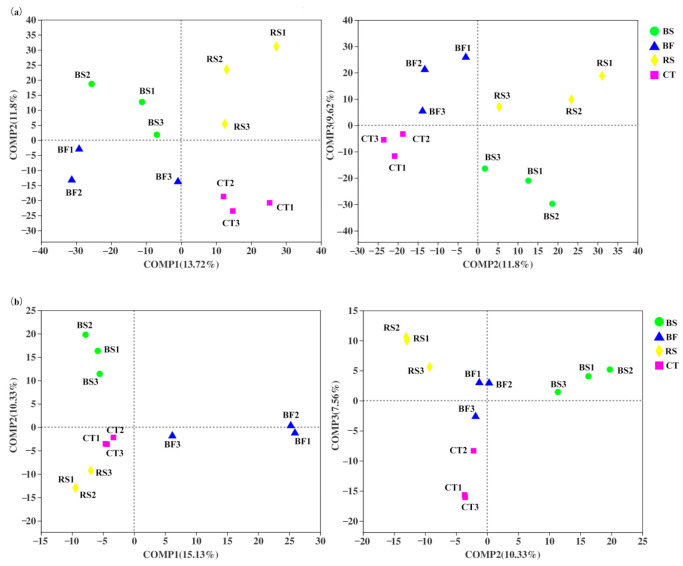
Partial Least Squares Discriminant Analysis score plots of bacterial (**a**) and fungal (**b**) communities for different soil and water conservation measures in the three soil depth layers.

**Figure 4 microorganisms-09-00319-f004:**
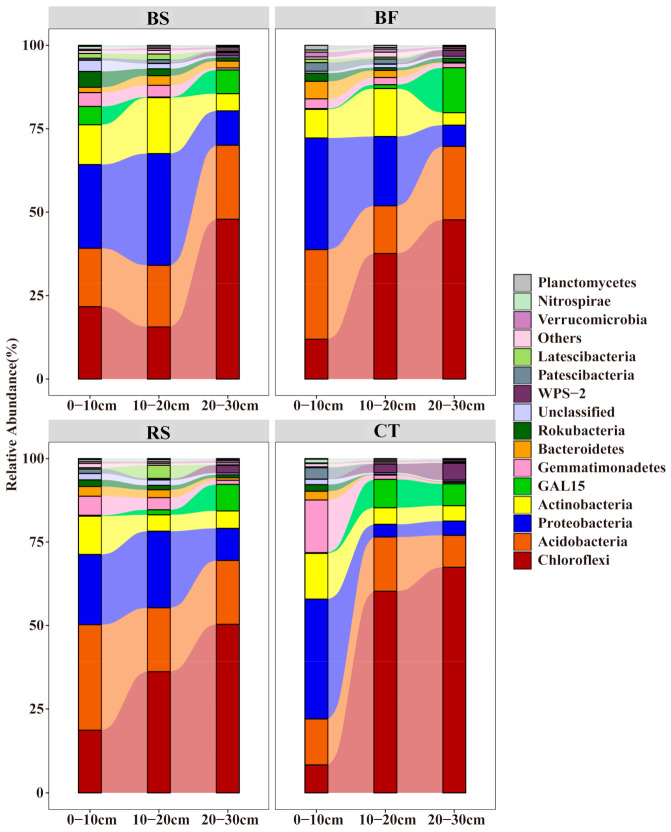
Phylum-level taxonomic composition of the bacterial community under different soil and water conservation measures. Note: the parts with an average abundance of less than 1% were merged and indicated by others in the figure.

**Figure 5 microorganisms-09-00319-f005:**
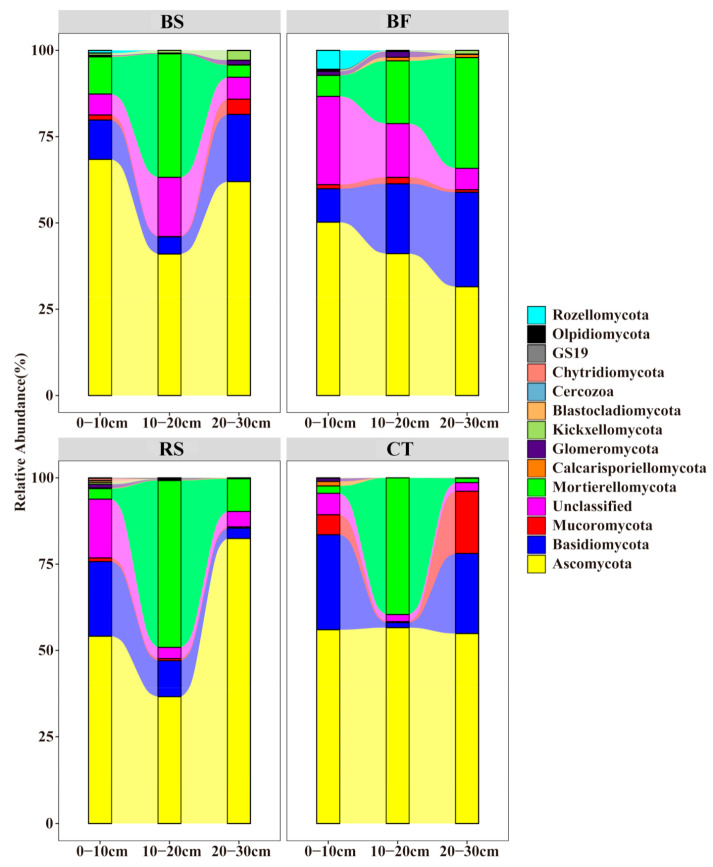
Phylum-level taxonomic composition of the fungal community under different soil and water conservation measures. Note: the parts with an average abundance of less than 1% were merged and indicated by others in the figure.

**Figure 6 microorganisms-09-00319-f006:**
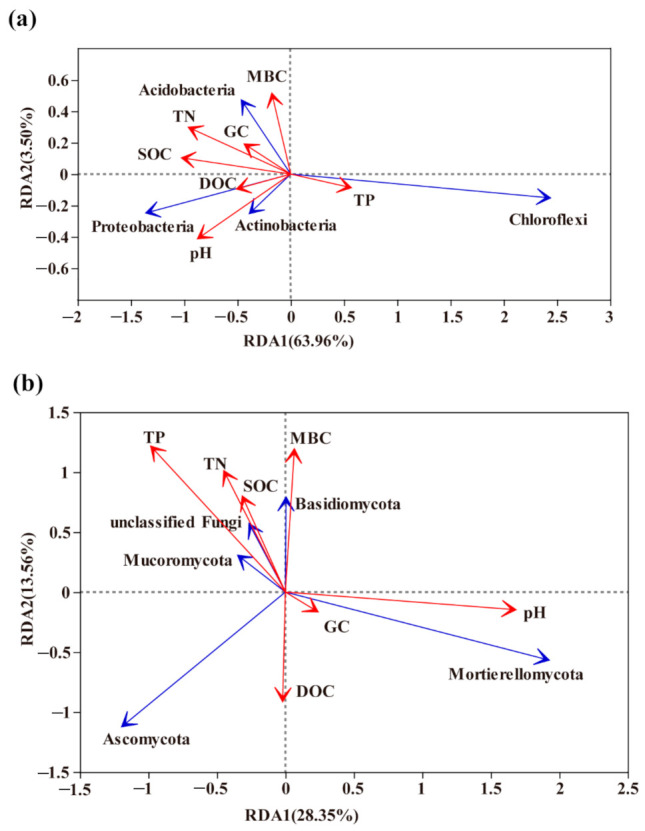
Redundancy Analysis (RDA) ordinate plots show the relationships among bacteria (**a**) and fungal (**b**) phyla (**blue** arrow) and soil properties (**red** arrow). Arrows indicate the direction and magnitude of variables.

**Figure 7 microorganisms-09-00319-f007:**
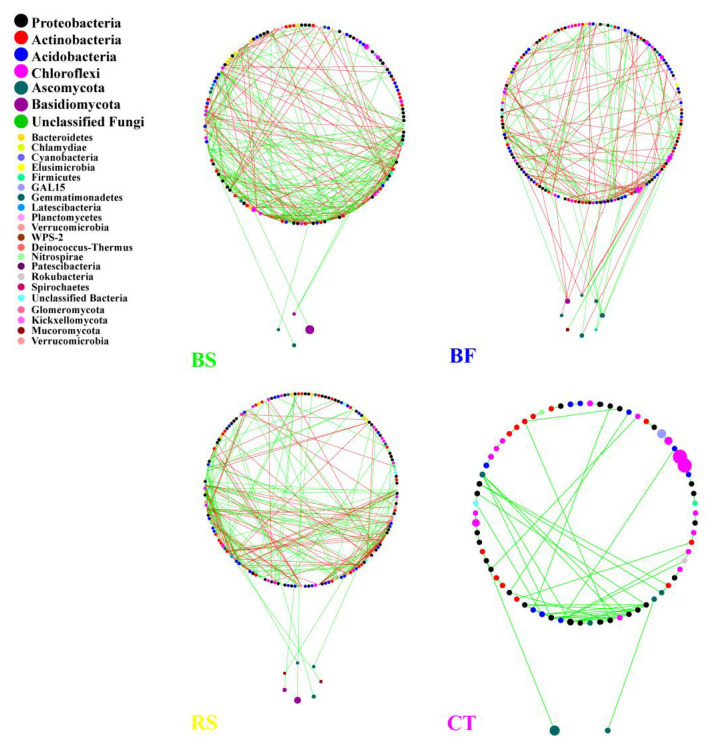
Molecular ecological networks of bacterial–fungal communities under different soil and water conservation measures. Green and red lines denote significant positive (Spearman correlation, *p* < 0.05, *r* > 0.8) and negative linear (Spearman correlation, *p* < 0.05, *r* < −0.8) relationships, respectively. The size of the circles represents the relative abundance of bacteria or fungi.

**Figure 8 microorganisms-09-00319-f008:**
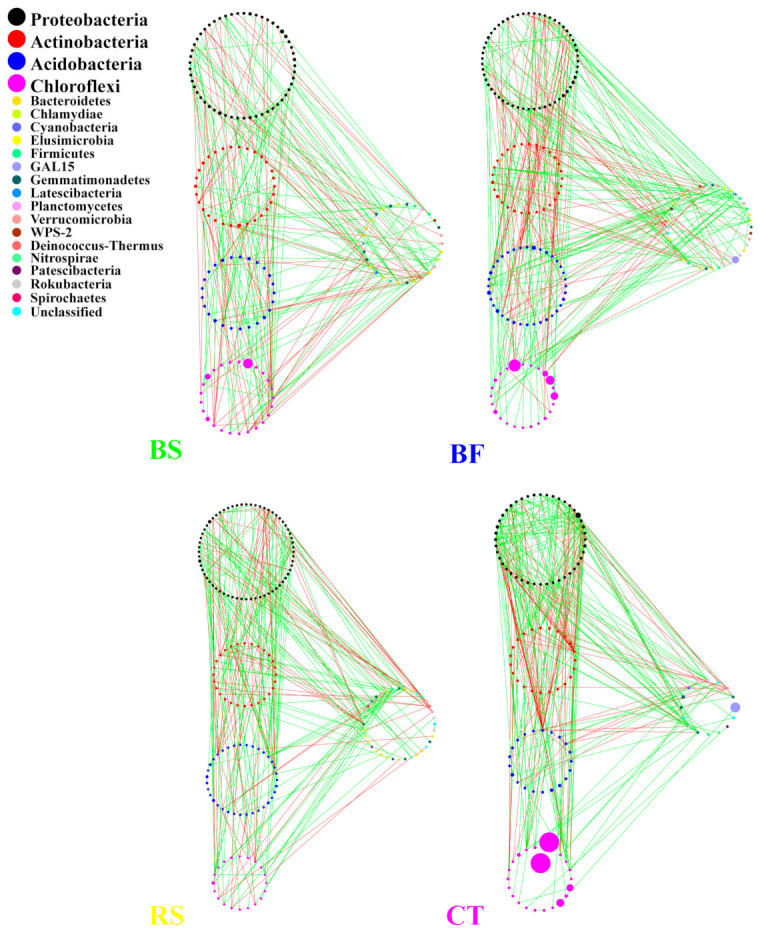
Molecular ecological networks of bacterial communities under different soil and water conservation measures. Green and red lines respectively denote significant positive (Spearman correlation, *p* < 0.05, *r* > 0.8) and negative linear (Spearman correlation, *p* < 0.05, *r* < −0.8) relationships. The size of the circles represents the relative abundance of bacteria.

**Figure 9 microorganisms-09-00319-f009:**
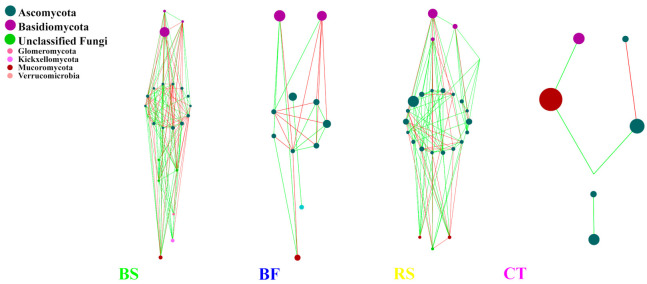
Molecular ecological networks of fungal communities under different soil and water conservation measures. Green and red lines respectively denote significant positive (Spearman correlation, *p* < 0.05, *r* > 0.8) and negative linear (Spearman correlation, *p* < 0.05, *r* < −0.8) relationships. The size of the circles represents the relative abundance of fungi.

**Table 1 microorganisms-09-00319-t001:** Effect of soil and water conservation measures on soil physicochemical properties and the bacterial and fungal quantities based on a 16S rRNA gene sequence assignment dataset with a 97% sequence similarity threshold.

Depths (cm)	Sample Name	TP (g kg^−1^)	TN (g kg^−1^)	pH	SOC (g kg^−1^)	GC (%)	DOC (mg kg^−1^)	MBC (mg kg^−1^)	Bacteria 16S rRNA	Fungal ITS rRNA
0–10	BS1	0.33 (0.02) ^bd^	1.30 (0.02)^c^	7.28 (0.04) ^b^	10.74 (1.17) ^e^	0.0093 (0.0025) ^ab^	28.70 (3.25) ^ad^	147.23 (44.46) ^a^	8.49 (0.02) ^a^	6.67 (0.02) ^e^
	BF1	0.32 (0.07) ^b^	1.44 (0.06)^c^	6.36 (1.22) ^b^	12.48 (0.60) ^c^	0.0071 (0.0024) ^ab^	29.53 (1.14) ^abc^	150.66 (76.16)^a^	8.80 (0.02) ^d^	6.58 (0.06) ^b^
	RS1	0.50 (0.04) ^c^	1.15 (0.12)^d^	6.93 (0.91) ^b^	9.79 (0.37) ^e^	0.0116 (0.0036) ^ab^	28.44 (5.05) ^ac^	132.29 (54.61) ^a^	8.45 (0.03) ^a^	6.36 (0.05) ^c^
	CT1	0.33 (0.03) ^b^	0.82 (0.03) ^e^	7.08 (0.13) ^bd^	7.69 (0.47) ^a^	0.0103 (0.0039) ^ab^	29.60 (0.58) ^a^	79.17 (43.42) ^ab^	8.40 (0.03) ^c^	6.27 (0.01) ^d^
10–20	BS2	0.38 (0.06) ^ad^	0.96 (0.08) ^b^	7.23 (0.09) ^b^	7.26 (0.66) ^ab^	0.0092 (0.0024) ^b^	30.42 (5.91) ^ad^	143.29 (69.01) ^a^	8.48 (0.02) ^a^	6.52 (0.03) ^f^
	BF2	0.29 (0.02) ^b^	0.94 (0.00) ^b^	7.06 (0.17) ^bd^	7.94 (0.04) ^b^	0.0086 (0.0007) ^b^	32.04 (3.89) ^a^	181.52 (33.36) ^a^	8.53 (0.05) ^ae^	6.81 (0.05) ^c^
	RS2	0.34 (0.00) ^ab^	0.85 (0.06)^b^	7.38 (0.19) ^b^	7.75 (0.06) ^b^	0.0132 (0.0030) ^b^	32.63 (2.89) ^ac^	114.53 (29.44) ^a^	8.73 (0.02) ^b^	6.67 (0.02) ^d^
	CT2	0.32 (0.05) ^ab^	0.71 (0.03) ^ae^	6.74 (0.33) ^d^	6.52 (0.51) ^a^	0.0052 (0.0026) ^a^	26.35 (6.70) ^a^	54.02 (11.38) ^b^	8.51 (0.04) ^a^	6.09 (0.00) ^b^
20–40	BS3	0.44 (0.21) ^abdef^	0.85 (0.07) ^b^	6.24 (0.65) ^a^	5.78 (0.72) ^d^	0.0094 (0.0016) ^ab^	33.66 (6.60) ^d^	150.40 (25.17) ^a^	8.44 (0.04) ^a^	6.30 (0.04) ^ab^
	BF3	0.38 (0.14) ^abef^	0.86 (0.11) ^b^	6.38 (0.65) ^ab^	6.06 (1.78) ^bd^	0.0095 (0.0018) ^ab^	25.37 (2.03) ^b^	186.92 (29.92) ^a^	8.59 (0.01) ^be^	6.37 (0.05) ^a^
	RS3	0.28 (0.02) ^e^	0.63 (0.02) ^a^	6.06 (0.78) ^ab^	4.90 (0.54) ^d^	0.0117 (0.0039) ^b^	32.64 (1.20) ^cd^	146.92 (17.18) ^a^	8.29 (0.03) ^c^	6.26 (0.04) ^b^
	CT3	0.85 (0.40) ^f^	0.63 (0.11) ^a^	5.39 (0.14) ^a^	4.66 (0.62) ^d^	0.0068 (0.0025) ^a^	22.18 (0.27) ^ab^	97.92 (19.16) ^b^	8.62 (0.01) ^b^	6.67 (0.04) ^c^

1, 2 and 3 represent soil sampling layers 0–10 cm, 10–20 cm and 20–40 cm (the same below); TP, total phosphorus (g kg^−1^); TN, total nitrogen (g kg^−1^); SOC, soil organic carbon (g kg^−1^); GC, gravel content (%); DOC, dissolved organic carbon (mg kg^−1^); MBC, microbial biomass carbon (mg kg^−1^); Unit of bacterial 16S rRNA and fungal ITS rRNA are log10 copies per gram of dry soil. Each value is the mean of three replicates with standard errors in parentheses. Within each row different letters show statistical significance at (*p* < 0.05). One-way ANOVA was performed on four soil and water conservation measures (*n* = 4) and three soil layers (*n* = 3).

**Table 2 microorganisms-09-00319-t002:** Pearson correlations between bacterial 16S rRNA gene copy number and soil properties under different soil and water conservation measures. Asterisks indicates significant correlation (** *p* < 0.01, * *p* < 0.05).

		TP	TN	pH	SOC	GC	DOC	MBC
measures	BS	0.44	0.46	0.36	0.47	−0.06	0.15	0.20
	BF	−0.03	0.88 **	−0.34	0.74 *	−0.55	−0.06	−0.54
	RS	0.17	0.33	0.70 *	0.52	0.27	0.11	−0.40
	CT	0.68 *	−0.81 **	−0.86 **	−0.93 **	−0.47	−0.62	0.45
layers	Surface	−0.39	0.72 **	−0.41	0.79 **	−0.51	0.14	0.33
	Middle	−0.001	−0.13	0.46	0.34	0.49	0.24	0.09
	Bottom	0.60 *	0.17	−0.19	0.03	−0.5	−0.70 *	−0.19

**Table 3 microorganisms-09-00319-t003:** Pearson correlations between fungal ITS rRNA gene copy number and soil properties under different soil and water conservation measures. Asterisks indicates significant correlation (** *p* < 0.01, * *p* < 0.05).

		TP	TN	pH	SOC	GC	DOC	MBC
measures	BS	−0.42	0.92 **	0.81 **	0.92 **	−0.08	−0.46	−0.13
	BF	−0.30	0.11	0.28	0.29	−0.03	0.79 *	0.11
	RS	0.04	0.22	0.66	0.41	0.27	0.05	−0.30
	CT	0.84 **	−0.50	−0.86 **	−0.73 *	0.17	−0.46	0.62
layers	Surface	−0.38	0.82 **	−0.05	0.77 **	−0.22	−0.13	0.57
	Middle	−0.14	0.76 **	0.58	0.84 **	0.65 *	0.51	0.77 **
	Bottom	0.72 **	−0.33	−0.50	−0.29	−0.56	−0.82 **	−0.62 *

**Table 4 microorganisms-09-00319-t004:** Parameters of interaction networks of bacterial–fungal communities under different soil and water conservation measures.

Soil Conservation Measures	Interaction Networks									
	Nodes	Links	Negative Links	Positive Links	Negative/ Positive	Average Geodesic Distance	Average Clustering Coefficient	Network Diameter	Average Connectivity	Modularity
BS	165	294	110	184	0.600	2.40	0.040	7	3.56	6
BF	166	192	95	97	0.979	2.56	0.286	7	2.31	16
RS	184	264	103	161	0.640	2.34	0.158	7	2.87	8
CT	72	61	0	61	0	1.70	0.153	4	1.69	3

**Table 5 microorganisms-09-00319-t005:** Parameters of interaction networks of bacterial communities under different soil and water conservation measures.

Soil Conservation Measures	Interaction Networks									
	Nodes	Links	Negative Links	Positive Links	Negative/ Positive	Average Geodesic Distance	Average Clustering Coefficient	Network Diameter	Average Connectivity	Modularity
BS	163	245	93	152	0.612	2.63	0.121	7	3.01	6
BF	186	283	100	183	0.546	2.95	0.348	11	3.04	15
RS	183	266	95	171	0.556	2.08	0.230	6	2.91	8
CT	126	304	103	201	0.512	2.23	0.291	6	4.83	8

**Table 6 microorganisms-09-00319-t006:** Parameters of interaction networks of fungal communities under different soil and water conservation measures.

Soil Conservation Measures	Interaction Networks									
	Nodes	Links	Negative Links	Positive Links	Negative/ Positive	Average Geodesic Distance	Average Clustering Coefficient	Network Diameter	Average Connectivity	Modularity
BS	23	131	54	77	0.701	1.648	0.769	5	11.391	3
BF	11	23	11	12	0.917	1.378	0.718	2	4.182	2
RS	25	99	31	68	0.456	1.957	0.644	4	7.92	2
CT	7	5	1	4	0.250	1.909	0.000	4	1.429	0

## Data Availability

The data presented in this study are available on request from the corresponding author.

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
