# Peer review of "Influence of Soil and Water Conservation Measures on Soil Microbial Communities in a Citrus Orchard of Southeast China"

_microorganisms, 2021, doi:10.3390/microorganisms9020319_

Round 1

Reviewer 1 Report

This is an interesting manuscript nevertheless it has got a few drawbacks.

Major issues:

  1. The most important issue I see is the fact that due to costly metagenomic analyses the replicates were mixed into single sample. I can understand that, however this significantly disabled performing statistical analyses. Thus in Figures 3, 5, 6 and other we can not know whether we see trends or just random differences. Thus I recommend trying estimation of the metagenomic data uncertainites and incorporate them into the calculations, tables, graphs etc.
  2. The discussion is quite long and rather speculative and the hypothesis stated in the aims is drawn in the text. I can recommend abbreviating of this part and structure it to more sub-sections. Also it would be interesting to link more the key properties (especially the nutrient cycling and soil improving) of the found taxa to the ecosystem functions.
  3. I can recommend moving some of the supplementary figures and tables into standard text. MDPI does not limit the length of the manuscript and it would significantly improve the understanding of the text.
  4. You use parametric statistical tests. This needs to be accomanied by a normality test (normality is important prerequisite,otherwise the conclusions are not valid and non-parametrical tests must be used).
  5. Figure 4 - The axis explain very limited variability. I can recommend to add other axes and discuss this lack of explanation.

Minor issues and recommendations:

  1. The abbreviations of the variants are not clear and are not related to them. I can recommend renaming them to something more intuitive.
  2. Figure 1 would be more clear in a form of table.
  3. Figure 1 - why sometimes a-labeled (or A-labeled) variants are the lowest and sometimes the highest?
  4. Section 3.2 - I am affraid part of the caption is missing.
  5. Figure 8 - justify, why you chose 0.8 for significance of the relationships?

Reviewer 2 Report

The manuscript entitled “Influence of soil and water conservation measures on soil microbial communities in a citrus orchard of southeast China” comprises the necessary updates of scientific novelty. Well written and summarized.

  • In the introduction, lines 73-80: This content can be used in the discussion and re-arrange the introduction.
  • Figure 1. Values … are significantly different. Values within … are significantly different. It's more confusing, let's make either Capital or smaller letters.
  • Results and discussion were well interpreted with figures and tables.

I would recommend the publication of this manuscript after addressing minor changes.

Round 2

Reviewer 1 Report

I generally agree with the corrections made nevertheless I have further comments to those particular points:

1. I can not agree with not working with uncertainties. One value has no meaning, you can not distinguish between trend and random results. Each analytical method (including the genetic methods) has it uncertainties which can be at least estimated, for example from the most uncertain procedure steps or use the published literature values. Methods like Monte-Carlo can be used to artifically introduce variability- Also your explanation of this point needss to be incorporated into the text.   

4. The results of the normality test shall be presented (e.g. as a supplementary material). Also now you mix together the parametric statistics (ANOVA) and non-parametric tests (Kruskal-Walilis test or Spearman correlation) which do not require the normally distribured data.

5. Adding third axis did not neccesirelly mean to have a 3D chart, these could be also two 2D charts (axis 1 vers. axis 2 and axis 2 vers. axis 3). The 3D chat is not bad but it requires additional helping lines for orientation of the data points.

e. Justification of chosing 0,8 needs to be incorporated into the text. However what is "strong" correlation is dependent also on the number of data points, it can be hardly arbitrarily chosen.
